# Extracellular Vesicle Treatment Alleviates Neurodevelopmental and Neurodegenerative Pathology in Cortical Spheroid Model of Down Syndrome

**DOI:** 10.3390/ijms24043477

**Published:** 2023-02-09

**Authors:** Natalie Baker Campbell, Yesha Patel, Tara L. Moore, Maria Medalla, Ella Zeldich

**Affiliations:** 1Department of Anatomy & Neurobiology, Boston University Chobanian & Avedesian School of Medicine, Boston University, Boston, MA 02118, USA; 2Commonwealth Honors College, University of Massachusetts Amherst, Amherst, MA 01003, USA; 3Center for Systems Neuroscience, Boston University, Boston, MA 02115, USA

**Keywords:** Down syndrome, Alzheimer’s disease, brain organoids, extracellular vesicles, mesenchymal stem cells, trisomy

## Abstract

Down syndrome (DS), or trisomy 21, is manifested in a variety of anatomical and cellular abnormalities resulting in intellectual deficits and early onset of Alzheimer’s disease (AD) with no effective treatments available to alleviate the pathologies associated with the disorder. The therapeutic potential of extracellular vesicles (EVs) has emerged recently in relation to various neurological conditions. We have previously demonstrated the therapeutic efficacy of mesenchymal stromal cell-derived EVs (MSC-EVs) in cellular and functional recovery in a rhesus monkey model of cortical injury. In the current study, we evaluated the therapeutic effect of MSC-EVs in a cortical spheroid (CS) model of DS generated from patient-derived induced pluripotent stem cells (iPSCs). Compared to euploid controls, trisomic CS display smaller size, deficient neurogenesis, and AD-related pathological features, such as enhanced cell death and depositions of amyloid beta (Aβ) and hyperphosphorylated tau (p-tau). EV-treated trisomic CS demonstrated preserved size, partial rescue in the production of neurons, significantly decreased levels of Aβ and p-tau, and a reduction in the extent of cell death as compared to the untreated trisomic CS. Together, these results show the efficacy of EVs in mitigating DS and AD-related cellular phenotypes and pathological depositions in human CS.

## 1. Introduction

Down syndrome (DS) is a genetic developmental disorder resulting from the triplication of the human chromosome 21 (HSA21) that affects approximately 1 in 700 live births [1,2,3]. Phenotypical characteristics of DS include craniofacial abnormalities, short stature, muscle hypotonia, and congenital heart defects (reviewed in [4]). DS is the most common form of intellectual disability, with a wide range of cognitive deficits [4], and people with DS are also at a higher risk for developing Alzheimer’s Disease (AD) at an earlier age than the general population [4,5].

On the anatomical level, the DS human brain demonstrates changes in cortical volume and abnormal grey and white matter development [6,7,8]. The overall reduced cortical volume [7] is accompanied by particularly small sizes of the cerebellum [7] and hippocampus [8], as well as a smaller volume of the frontal and temporal lobes [9,10]. Decreased cortical size has been attributed to a diminished surface area in the affected regions [11] and is associated with aberrant cortical lamination, altered axonal and dendritic arborization, deficient synaptogenesis [12,13], and hypocellularity [14,15,16,17,18,19,20]. The hypocellularity, represented by a reduction in neuronal numbers, is found in the neocortex, hippocampus, dentate gyrus, and cerebellum in fetal DS brains during the second trimester [14,15,16,17,18,19,20,21]. Furthermore, these structural differences are also observed in mouse models of DS. In the most widely used mouse model, Ts65Dn, the hippocampal hypocellularity [22], specifically in the dentate gyrus [23], has been linked to learning, memory, and behavior deficits [24]. The cognitive deficits observed in DS are further attributed to a disturbance in the white matter. Aberrant myelin biology in DS is manifested in delayed and diminished myelination [25,26] resulting in fewer myelinated axons with thinner myelin sheaths [27].

The hypocellularity associated with DS and the reduction in neuronal numbers is partially attributed to a diminished proliferating capacity of neural progenitor cells (NPCs) as observed in fetal DS human brain tissue (reviewed in [15,28,29,30,31]) and in cellular models of DS [32,33]. Decreased numbers of neurons are also linked to diminished neuronal output and abnormal cortical lamination [12,31,34]. These changes have been recapitulated recently in DS-derived three-dimensional cellular model organoids. Studies from our laboratory and others [33,35] demonstrated deficient detection of the neurons populating deep and superficial cortical layers in trisomic organoids. Furthermore, using single-cell transcriptomic studies, we found a particular vulnerability of trisomic excitatory neurons, transcriptomically corresponding to cortical layer IV [35], contributing further to the aberrant cortical development in DS.

The abnormal neurogenesis observed in DS is also attributed to a precocious gliogenic shift causing neural progenitor cells to switch earlier to the production of glial cells at the expense of neurons (reviewed in [36,37]). In fact, fetal DS brains and DS-derived iPSCs show increased glial fibrillary acidic protein (GFAP) expression, suggesting higher astrocyte content compared to euploid controls [37,38,39]. Increased numbers of astrocytes linked to a trisomy have been shown to have negative effects on neuron survival by affecting neuron ion channel maturation and neurite outgrowth, causing synaptic dysfunction, and changing the microenvironment observed in DS brains [38].

The gene dosage effect of *Amyloid precursor protein* (*APP*), one of the HAS21 genes, is considered one of the primary reasons for the comorbidity between DS and AD (DS-AD) [40]. AD is characterized by amyloid β (Aβ) plaque formation, hyperphosphorylation of tau (p-tau), neurofibrillary tangles, and increased neuronal cell death. Normalization of the gene dosage of *APP* in a mouse model does reduce the accumulation of plaques and diminish the levels of Aβ proteins, but does not have an impact on tau hyperphosphorylation, neurofibrillary tangle (NFT) abundance, or neuronal cell death [41]. These data suggest that there are other mechanisms in action in brains with DS that cause AD pathology [41]. However, there is also evidence to suggest that when *APP* is triplicated without the triplication of the entire HSA21, it is sufficient to induce early-onset AD pathology [42]. Accumulation of p-tau is also observed in DS brain tissue and is correlated with cognitive decline, increasing burden of Aβ depositions, and accelerated neurodegeneration [43]. However, as of today, there is no cure for AD and DS-AD, and clinical trials for potential AD therapeutics had a 99.6% failure rate between 2002 and 2012 [44].

Extracellular vesicles (EVs) (or exosomes) are membrane-bound vesicles released from cells into the extracellular matrix as a form of intracellular communication [45]. EVs are secreted by a wide range of cell types [46] and contain miRNA, mRNA, small fragments of DNA, lipids, growth factors, cytokines, and other cellular components [47,48,49]. Due to their ability to transport cargo from one cell to another, EVs have recently emerged as a potential therapeutic strategy to treat a variety of diseases, including neurological disorders (reviewed in [45]).

We have recently shown that EVs exhibit therapeutic effects following cortical injury in middle-aged rhesus monkeys [50]. In our model, EVs were isolated from mesenchymal stromal cells (MSCs) derived from the bone marrow of a young healthy rhesus monkey and administered intravenously post-injury. Monkeys that received the EV treatment exhibited greater recovery of movement on a fine motor grasping task and were able to return to their baseline function more rapidly and with more precision than the control group. On the cellular level, we found that EVs reduced excitotoxicity [51], aided in the shift of microglia from pro-inflammatory to anti-inflammatory phenotypes [52], and enhanced remyelination [53] and neuronal plasticity [51] in the brain tissue of the EV-treated subjects. These studies identified the therapeutic potential of EVs to mitigate injury-related impairment and promote functional recovery. 

The pleiotropic effects of EVs could be explored as a possible therapeutic for AD and DS-AD. However, it is not possible to study AD pathology in non-human primates since, although monkey models do experience normal aging and cognitive decline [54], they do not exhibit the hallmarks of AD pathology, such as the accumulation of Aβ proteins and the hyperphosphorylation of tau [55]. In addition, there are only a few described cases of trisomy (trisomy 22) in non-human primates [56,57]. Therefore, these potential therapeutic properties of EVs in relation to AD and DS need to be studied in different model systems.

In the current study, we used induced pluripotent stem cell (iPSC) isogenic lines derived from human patients with DS to generate cortical spheroids (CS) (or organoids) following a published protocol [58]. These CS contain several types of neural progenitor cells and excitatory neurons as well as inhibitory neurons, oligodendrocytes, and astrocytes [35] and recapitulate the developmental trajectory of an embryonic brain at mid-gestation [58,59].

DS-patient-derived CS are especially useful for exploring therapeutic approaches because we can modulate cellular and molecular phenotypes related to the disorder in a comprehensive human-derived physiological environment and explore the effectiveness of potential therapeutics on DS and AD pathological features. Using CS generated from isogenic euploid and trisomic cell lines, we demonstrated that administration of MSC-EVs in trisomic CS preserved the size of the spheroids, promoted neurogenesis, diminished cell death, and reduced the depositions Aβ and p-tau, suggesting a novel therapeutic approach to mitigate cellular phenotypes in DS.

## 2. Results

In the present study, we examined the potential efficacy of MSC-EVs in DS-related neurodevelopmental cellular phenotypes as well as neurodegenerative AD-related pathology in the human cellular model. For this purpose, we generated CS from isogenic paired lines containing a trisomic WC-24-02-DS-M line, and a euploid cell line, WC-24-02-DS-B, derived from a female with DS using the method adapted from [58] (Figure 1A). Both cell lines underwent three independent differentiations into CS. EV treatment was applied to trisomic CS on days 80, 87, 94, 101, 108, and 115, and CS were analyzed on day 120, as shown in Figure 1, illustrating our experimental design. Day 120 was chosen by us since at this developmental stage CS express markers of deep and superficial layer neurons [35], contain mature glial cells [58], and display AD-related pathological features [60]. The CS generated in our laboratory for this study contained neurons, oligodendrocytes, and astrocytes, and displayed proliferative zones surrounding lumen-like structures. 

### 2.1. EV Treatment Preserved the Size of Trisomic CS

Reduced cortical volume is one of the DS pathological features and has been demonstrated in trisomic human brain tissue and mouse models of DS [10,61] as well as in iPSC-derived models of DS [21,32,33]. Cortical thinning is also a common pathological feature observed in DS and AD [62,63]. In line with our previously published study [35], there were no significant differences in trisomic CS diameter after day 30 of differentiation when compared to euploid CS (Figure 1B). However, at later time points, trisomic CS displayed significantly reduced diameters compared to euploid CS when measured on day 50 (2475 ± 48.21 µm, euploid; 2112 ± 36.72 µm, trisomic; *p* < 0.0001, one-way ANOVA), day 80 (2380 ± 22.22 µm, euploid; 2061 ± 21.23 µm, trisomic; *p* < 0.0001, one-way ANOVA) and day 120 (2388 ± 79.41 µm, euploid; 1952 ± 27.07 µm, trisomic; *p* < 0.0001, one-way ANOVA, Figure 1B). On day 120, we also assessed the difference in diameter between untreated trisomic CS and trisomic CS that were treated with EVs (referred to hereinafter as trisomic EV+) starting on day 80. The media changes of the euploid and trisomic controls were synchronized with the EV treatments applied weekly to the trisomic EV+ group. The day 80 timepoint was chosen for two reasons. First, AD pathology has been shown to be present in 3D cortical spheroids as early as day 110 [60] and we aimed to determine whether EV treatment could interrupt the development of Aβ depositions and p-tau. Second, superficial layer neurons are found in spheroids after day 100 and are associated with indirect neurogenesis observed in the gyrencephalic brain [64]. In addition, based on the protocol we applied [58], markers of oligodendrocytes lineage such as PLP1, and MBP were fully detected on week 14 (day 98). Treating the CS with EVs starting on day 80 provided us with an opportunity to assess their efficacy during more advanced stages of neurogenesis and gliogenesis. On day 120, both EV-treated and untreated trisomic CS were significantly smaller in diameter than euploid control CS (2388 ± 79.41 µm, euploid; 1952 ± 27.07 µm, trisomic; 2122 ± 44.49 µm, trisomic EV+; *p* < 0.0001, one-way ANOVA, Figure 1C). However, the diameter of the trisomic EV+ CS was significantly larger when compared to untreated trisomic CS, showing that EV administration partially preserved the volume of trisomic CS (1952 ± 27.07 µm, trisomic; 2122 ± 44.49 µm, trisomic EV+, *p* = 0.0009, one-way ANOVA, Figure 1C). With this finding, we investigated the potential processes that could account for this EV-related preservation of trisomic CS volume, such as proliferation, neurogenesis, and gliogenesis, as well as processes associated with cell viability and death.

### 2.2. EV Treatment Did Not Affect the Proliferation of Neural Progenitors (NPCs) within Trisomic CS

It has been shown in both human fetuses and mouse models of DS that there is an abnormal proliferation of neural progenitors in trisomy, resulting in hypocellularity of different brain regions in DS [15,65]. In order to investigate potential changes in the population and proliferation of NPCs between euploid and trisomic CS, as well as the possible effect of the EVs on trisomic NPCs, we focused on the cells populating the ventricular zone (VZ)-like areas that were enriched in proliferating NPCs and could be identified visually based on their morphological appearance and enrichment with the proliferative marker Ki67 as well as with the SOX2 marker specific to progenitor cells (Figure 2A). Thus, we quantified the expression of Ki67 positive (Ki67+) cells to measure the differences in the proliferation of NPCs and the expression of SOX2 positive (SOX2+) cells to elucidate the effects of trisomy and EV treatment on the numbers of NPCs (Figure 2A). No significant differences were found in the percentages of Ki67+ cells (100.0 ± 8.311%, euploid; 127.2 ± 13.89%, trisomic; *p* = 0.1012) or SOX2+ cells (100.0 ± 9.561%, euploid; 112.3 ± 14.10%, trisomic; *p* = 0.4747) between NPCs populating VZ-like structures in euploid and trisomic CS (Figure 2B,C). These results suggest that the previously observed differences in developmental trajectories of neurons in trisomic CS [35] do not originate from alterations in the proliferation capacity of trisomic NPCs. We compared trisomic untreated and trisomic EV+ CS and also found no significant differences in the percentages of Ki67+ (127.2 ± 13.89%, trisomic; 130.0 ± 19.99%, trisomic EV+; *p* = 0.9101) and SOX2+ cells (112.3 ± 14.10%, trisomic; 101.8 ± 12.41%, trisomic EV+; *p* = 0.5801). Consequently, we also did not observe any significant differences between trisomic EV+ CS and euploid CS with regard to the percentages of Ki67+ (100.0 ± 8.311%, euploid; 130.0 ± 19.99%, trisomic EV+; *p* = 0.1754) and SOX2+ cells (100.0 ± 9.561%, euploid; 101.8 ± 12.41%, trisomic EV+; *p* = 0.9073). Our findings indicate no changes in the proliferation of trisomic NPCs in our cellular system and suggest that the observed differences in the size of trisomic CS (both due to trisomy and following the administration of EVs) might be attributed to the later developmental stages, such as neurogenesis or gliogenesis, or enhanced degenerative processes occurring within trisomic CS. 

### 2.3. Treatment with EVs Can Ameliorate Aberrant Neurogenesis of Specific Neuronal Subpopulations in Trisomic CS

To evaluate the effect of EVs on neurogenesis, we stained CS with special AT-rich sequence-binding protein 2 (SATB2), a marker of superficial neurons, COUP-TF-interacting protein 2 (CTIP2), which marks deeper layer neurons, and T-Box Brain Transcription Factor 1 (TBR1), which stains newly formed neurons (Figure 3A). 

In line with the data previously published by others [21,65] and from our group [35], there was a significant decrease in the percentage of SATB2+ cells in trisomic CS as compared to euploid counterparts (100.0 ± 9.364% euploid; 70.22 ± 9.443%, trisomic; *p* = 0.0283) (Figure 3B). Treatment with EVs resulted in a trend towards an increase in the percentage of SATB2+ cells in trisomic EV+ CS compared to untreated trisomic CS (70.22 ± 9.443%, trisomic; 118.4 ± 22.80%, trisomic EV+; *p* = 0.0606), suggesting that EVs could promote the generation of superficial layer neurons. Noticeably, the percentage of SATB2+ cells in trisomic EV+ CS did not significantly differ from the percentage of SATB2+ cells in euploid CS (100.0 ± 9.364% euploid; 118.4 ± 22.80%, trisomic EV+; *p* = 0.4621), indicating that the administration of EVs ameliorated the reduction in neurogenesis associated with trisomy and facilitated a return to the levels seen in euploid CS (Figure 3B). 

As expected from the previous observations [35], the percentage of deep layer neurons, as marked with CTIP2, was significantly lower in trisomic CS compared to euploid CS (100.0 ± 12.00% euploid; 66.71 ± 11.22%, trisomic; *p* = 0.0466) (Figure 3C). Treatment with EVs did not reach statistical significance when comparing trisomic CS to trisomic EV+ CS (66.71 ± 11.22%, trisomic;122.6 ± 37.69%, trisomic EV+; *p* = 0.1678). However, the percentage of CTIP2+ cells in trisomic EV+ CS was no longer significantly different when compared to euploid CS (100.0 ± 12.00% euploid; 122.6 ± 37.69%, trisomic EV+; *p* = 0.5731), suggesting a partial rescue of neurogenesis in trisomic EV+ CS and a partial return to the levels demonstrated in euploid CS (Figure 3C).

Similar to SATB2+ and CTIP2+ cells, there was a significant decrease in the percentage of TBR1+ cells in trisomic CS when compared to euploid (100.0 ± 4.837%, euploid; 56.80 ± 7.253%, trisomic; *p* < 0.0001) (Figure 3D), demonstrating once again the abnormal neurogenesis associated with trisomy. However, the percentage of TBR1+ cells was also lower in trisomic EV+ CS compared to trisomic untreated samples (56.80 ± 7.253%, trisomic; 56.96 ± 9.371%, trisomic EV+; *p* = 0.9895), revealing no effect of EV treatment on the rescue of TBR1+ neurons (Figure 3D). Taken together, these data show that DS is associated with a reduction in neurogenesis, and EV treatment can partially ameliorate the reduction of specific subpopulations of neurons.

### 2.4. EV Treatment Modified GFAP and CC1 Expression in Trisomic CS

After addressing neuronal proliferation and differentiation in CS, we measured markers of astrocytes and oligodendrocytes to assess the effects of trisomy and EV treatment on glial cells. Precocious gliogenic switch has been implicated in DS, resulting in an increased population of astrocytes found in DS human and mouse brains [39,66,67,68].

To assess whether the increased production of astrocytes is detected in trisomic CS, we stained for GFAP, an astrocytic marker [69]. We also evaluated the possible implications of EV administration on gliogenesis in trisomic CS (Figure 4A). We detected significantly greater expression of GFAP in untreated trisomic CS compared to euploid CS (100.0 ± 21.74%, euploid; 877.7 ± 76.13%, trisomic; *p* < 0.0001), supporting the increased production of GFAP-positive cells in trisomic CS. GFAP expression was significantly reduced by EV treatment compared to untreated trisomic CS (877.7 ± 76.13%, trisomic; 339.5 ± 100.2%, trisomic EV+; *p* = 0.0001). However, the immunoreactivity of GFAP in trisomic EV+ CS was still significantly elevated compared to euploid CS (100.0 ± 21.74%, euploid; 339.5 ± 100.2%, trisomic EV+; *p* = 0.0274; Figure 4B). Nevertheless, these data suggest that EV treatment can affect the generation of astrocytes within trisomic CS and implicates the potential effect of the gliogenic switch previously described in trisomy. 

Gliogeneic shift associated with DS can also affect the production of oligodendrocytes, leading to change in their cell fate acquisition [68,70] as well as abnormal oligodendrocyte differentiation and deficient myelin production [25,27]. Therefore, we also measured the expression of anti-adenomatous polyposis coli clone CC1 (CC1), a marker of oligodendrocytes [71]. We observed significantly less expression of CC1 in trisomic CS compared to euploid (100.0 ± 12.77%, euploid; 63.09 ± 12.15%, trisomic; *p* = 0.0450). EV treatment did not have an effect on the proportion of oligodendrocytes in the trisomic CS (63.09 ± 12.15%, trisomic; 60.17 ± 9.323%, trisomic EV+; *p* = 0.8504). There were still significantly fewer oligodendrocytes in trisomic EV+ CS compared to euploid CS (100.0 ± 12.77%, euploid; 60.17 ± 9.323%, trisomic EV+; *p* = 0.0180). These results elucidate the effect of trisomy on oligodendrocytes in CS and demonstrate that EVs have no effect on the reduction in CC1 cells in trisomic CS.

### 2.5. EV Treatment Decreased Cell Death in Trisomic CS

While volume loss in trisomic CS can be due to diminished generation of neurons, it can also be due to increased cell death. To assess the role of cell death in CS development and the effect of EV treatment on this process, we stained for cleaved caspase-3 (CC3), a marker of programmed cell death (Figure 5A). CC3 staining has been widely used as an indicator of cell death in organoids and spheroids [72,73,74,75]. In line with our previous data, untreated trisomic CS showed a significant increase in the extent of CC3 detection when compared to euploid controls (100.0 ± 13.19%, euploid; 202.1 ± 34.77%, trisomic; *p* = 0.0106), indicating enhanced cell death in trisomic CS (Figure 5B). The expression of CC3 was significantly reduced in trisomic EV+ CS compared to untreated trisomic CS (202.1 ± 34.77%, trisomic; 89.52 ± 14.99%, trisomic EV+; *p* = 0.0059), suggesting a neuroprotective effect induced by EV treatment (Figure 5B). CC3 expression in trisomic EV+ CS was not significantly different from euploid controls (100.0 ± 13.19%, euploid; 89.52 ± 14.99%, trisomic EV+; *p* = 0.6029), indicating rescue of cells from apoptotic processes induced by trisomy. 

### 2.6. EVs Alleviated AD Pathology in Trisomic CS

AD-related pathological depositions were previously reported to be detected in trisomic organoids on day 110 [60]. Therefore, we first examined whether trisomic EVs generated in our laboratory displayed AD-associated hallmarks: Aβ depositions and p-tau, and whether treatment with EVs can affect these pathological features of AD. Since the depositions of Aβ are detected in both soluble and insoluble fractions in DS brains [76] and the accumulation of Aβ is more profound in the insoluble fraction in DS and AD pathology [77], we comprehensively assessed the levels of Aβ40 and Aβ42 in soluble and insoluble fractions in euploid and trisomic untreated and EV-treated CS. For this purpose, the soluble and insoluble fractions of CS were separated and analyzed for both Aβ40 and Aβ42 concentrations using ELISA. 

The concentration of soluble Aβ40 was significantly greater in trisomic CS when compared to euploid controls (100.0 ± 8.883%, euploid; 177.4 ± 12.72%, trisomic; *p* < 0.0001; Figure 6A). Noticeably, EV treatment significantly reduced the levels of soluble Aβ40 in trisomic CS (177.4 ± 12.72%, trisomic; 129.7 ± 10.89%, trisomic EV+; *p* = 0.0084). We did not see any significant differences in the concentration of soluble Aβ42 between euploid and trisomic CS (100.0 ± 8.122%, euploid; 132.0 ± 34.06%, trisomic; *p* = 0.3834) or between trisomic and trisomic EV+ CS (132.0 ± 34.06%, trisomic, 85.27 ± 9.238 trisomic EV+; *p* = 0.2144; Figure 6B). Additionally, there was no significant difference between soluble Aβ42 concentrations in euploid and trisomic EV+ CS (100.0 ± 8.122%, euploid; 85.27 ± 9.238 trisomic EV+; *p* = 0.2465).

Our study revealed a significant increase in insoluble Aβ40 and Aβ42 concentrations in untreated trisomic CS when compared to euploid controls (insoluble Aβ40: 103.2 ± 7.173%, euploid; 181.6 ± 13.99%, trisomic; *p* < 0.0001; insoluble Aβ42: 100.0 ± 5.557%, euploid; 179.0 ± 15.31%, trisomic; *p* = 0.0005; Figure 6C,D). Remarkably, EV treatment resulted in a significant reduction in both insoluble Aβ40 and Aβ42 in trisomic CS (insoluble Aβ40: 181.6 ± 13.99%, trisomic; 119.9 ± 10.19%, trisomic EV+; *p* = 0.0016; insoluble Aβ42: 179.0 ± 15.31%, trisomic; 116.8 ± 9.172%, trisomic EV+; *p* = 0.0033). These concentrations of insoluble Aβ40 and Aβ42 found in trisomic EV+ CS were not significantly different from the levels in euploid CS (insoluble Aβ40: 103.2 ± 7.173%, euploid; 119.9 ± 10.19%, trisomic EV+; *p* = 0.1925; insoluble Aβ42: 100.0 ± 5.557%, euploid; 116.8 ± 9.172%, trisomic EV+; *p* = 0.1376), revealing the ability of EVs to ameliorate Aβ pathology in CS models of DS. 

The ratio of Aβ42/Aβ40 has recently become a more prevalent disease marker for AD pathology in clinical settings [78]. However, there was no significant change in the ratio of soluble or insoluble Aβ42/Aβ40 in trisomic CS compared with euploid (soluble Aβ42/Aβ40: 1.066 ± 0.1530, euploid; 0.8074 ± 0.2056, trisomic; *p* = 0.3286; insoluble Aβ42/Aβ40: 1.028 ± 0.0570, euploid; 0.8822 ± 0.0719, trisomic; *p* = 0.1324), nor was there a significant difference in trisomic EV+ CS when compared to trisomic (soluble Aβ42/Aβ40: 0.8074 ± 0.2056, trisomic;1.008 ± 0.1200, trisomic EV+; *p* = 0.4134; insoluble Aβ42/Aβ40: 0.8822 ± 0.0719, trisomic; 0.8695 ± 0.0.926; trisomic EV+; *p* = 0.9148; Figure 6E,F). Furthermore, there was no significant difference between euploid CS and trisomic EV+ CS in either fraction (soluble Aβ42/Aβ40: 1.066 ± 0.1530, euploid; 1.008 ± 0.1200, trisomic EV+; *p* = 0.7696; insoluble Aβ42/Aβ40: 1.028 ± 0.0570, euploid; 0.8695 ± 0.0.926; trisomic EV+; *p* = 0.1652; Figure 6E,F). These data show that EV treatment mitigates the depositions of Aβ40 in both fractions and the presence of Aβ42 in the insoluble fraction in trisomic CS but does not have an effect on the ratio of Aβ42/Aβ40 in either fraction. We were also able to observe the depositions of Aβ by staining CS with 4G8 antibody using IHC (Figure 6G,Gi)

Next, we evaluated the extent of p-tau at Serine 396 using the PHF13 antibody in CS whole homogenates (Figure 7A and Appendix A). We found that there was a trend towards an increase in tau phosphorylation in trisomic CS when compared to euploid (100.0 ± 6.101, euploid; 128.3 ± 12.10, trisomic; *p* = 0.0526) and there was a significant decrease in p-tau concentration in trisomic EV+ CS when compared to both euploid (100.0 ± 6.101, euploid; 49.21 ± 7.825, trisomic EV+; *p* = 0.0006) and untreated trisomic CS (128.3 ± 12.10, trisomic; 49.21 ± 7.825, trisomic EV+; *p* < 0.0001; Figure 7B). These results show that the administration of EVs was able to reduce the extent of p-tau in trisomic CS. We also observed p-tau in CS by staining with PHF13 antibody using IHC (Figure 7C,Ci). Noticeably, we also attempted to detect tau phosphorylation on Se202/Thr305 using the AT8 antibody, but the levels of p-tau in CS homogenates or in the CS slices using IHC were below detection with this antibody. Overall, these results indicate that EVs can decrease the extent of AD pathology by reducing depositions of Aβ and p-tau in trisomic CS. Altogether, our data suggest that EVs may mitigate pathological neurodevelopmental and neurodegenerative phenotypes related to trisomy.

## 3. Discussion

In this study, we used isogenic patient-derived iPSCs to generate euploid and trisomic CS to identify the effects of EVs on DS and DS-AD pathology. The results demonstrated that MSC-EVs have a therapeutic effect on cellular phenotypes and AD-related pathology in our 3D model of DS-derived CS. Our previous studies have shown that in vivo treatment with these MSC-EVs after cortical injury in aged rhesus monkeys facilitated recovery of fine motor function [50] through reducing excitotoxicity [51] and microglial inflammatory phenotypes [52] and enhancing myelination [53] and neuroplasticity [51]. Our data here extend these findings and support the therapeutic potential of MSC-EVs to ameliorate AD-related pathology in vivo.

Previous research showed that human adipose-derived EVs reduce Aβ levels, decrease microglia activation, rescue memory deficits, and promote neurogenesis when delivered intranasally in an APP/PS1 transgenic mouse model of AD [79]. Furthermore, EVs from healthy neural stem cells improved cell survival and proliferation in an ischemic stroke rat model [80]. Finally, EVs derived from dental pulp stem cells were shown to have anti-apoptotic effects in dopaminergic neurons [81], demonstrating the therapeutic potential of EVs in human cell models. However, to date, there have been no studies addressing the therapeutic effects of EVs on DS-AD-related pathology. 

To begin to fill this gap in the knowledge, we treated trisomic CS with EVs. At the neurodevelopmental level, EVs preserved the volume of trisomic CS and partially rescued neurogenesis-associated deficits in the production of deep and superficial layer neurons and the overproduction of astrocytes.

Reduced brain volume is observed as early as the fetal development stage in DS [12]. Brains of children with DS between birth and five years old are shown to have reduced densities and numbers of neurons [19]. Perturbations in neurogenesis have been observed in the cerebellum [15], hippocampus [31], inferior temporal gyrus [37], and cerebral cortex [20,82] of fetal DS brains. Furthermore, reductions in cortical volume and abnormal cortical lamination have been observed in fetuses with DS as early as gestational week 23 [12]. Similar hypocellularity has also been found in Ts65Dn mouse models of DS [29,65]. This hypocellularity has been partially attributed to the aberrant proliferation and reduced pool of neural progenitors in the DS brain. The reduction in the NPC population has been detected in the hippocampal dentate gyrus, as well as in the germinal matrix of the ventricles, in DS fetuses as early as 17–21 weeks of gestation [29,83]. This was also confirmed in the Ts65Dn mouse model [30] and DS-derived organoid models [33,35]. However, other studies found increased numbers of neural progenitors in the SVZ region in the medial ganglionic eminence in Ts65Dn and Ts1Cje mouse models, leading to the enhanced generation of inhibitory neurons accompanied by a paucity in the number of excitatory neurons [84,85]. In addition, impaired neurogenesis was attributed to a biphasic cell cycle defect manifested in the diminished proliferating capacity of NPCs in the early stages, followed by increased proliferation affecting the neurogenic phase of the neurogenesis, as was shown in a DS-derived iPSC-based model [84]. In the current study, the examination of NPCs revealed no difference in the proliferative capacity of NPCs and their numbers between euploid and trisomic CS. Consequently, EVs did not have an impact on the NPC population through the assessment of the markers tested in the current study. Further investigation might be needed to understand the mechanisms involved in the proliferative and cell cycle parameters of the NPCs in our CS model.

A deficient generation of deep and superficial cortical layer neurons has been recapitulated in DS-derived trisomic organoids in our laboratory [35] and others [33]. Furthermore, excitatory neurons corresponding transcriptomically to cortical layer IV neurons showed the most altered divergence in developmental trajectory between euploid and trisomic samples [35]. The deficient neurogenesis was shown to be mitigated by genetic editing and pharmacological inhibition targeting the DS cell adhesion molecule (DSCAM)/p21-activated kinase 1 (PAK1)-mediated pathway [33]. Our current study confirms the deficient generation of deep and superficial cortical layer neurons in trisomic CS. Noticeably, EV treatment of trisomic CS resulted in a partial rescue of neurogenesis, reflected in the increased neuronal numbers, comparable to the numbers in euploid CS, and more preserved size of the trisomic spheroids. 

Developmentally, neurogenesis is followed by gliogenesis and results in NPCs’ generation of oligodendrocytes and astrocytes. The shift between the neurogenesis and gliogenesis processes is abnormal in DS, causing cells to be disproportionately directed into a glial fate instead of neuronal (reviewed in [36]). Studies have shown that this shift can be attributed to the aberrant activation of the Sonic Hedgehog signaling pathway, leading to the generation of astrocytes at the expense of neurons and oligodendrocytes [68]. Administration of Sonic Hedgehog pathway Smoothened agonist (SAG) rescued neurogenesis in the cerebellum of Ts65Dn mice [86] and corrected the differentiation of transitioning NPC into cells committing to oligodendrocyte lineage [70]. Our results show increased generation of astrocytes in trisomic CS that was accompanied by decreased production of oligodendrocytes, marked with the general oligodendrocyte marker CC1. EV administration diminished the extent of GFAP staining, suggesting a potential correction of the overproduction of astrocytes in trisomic CS, with no effect on the population of oligodendrocytes. Interestingly, in vivo, we found that EV treatment after cortical injury showed no effect on oligodendrocyte precursor cells, but prevented damage in oligodendrocytes and enhanced myelin plasticity [53]. The reduction in the astrocyte population induced by EV treatment might also be related to the anti-inflammatory effects of EVs previously observed in microglia [52]. 

Our results suggest that (i) disrupted neurogenesis and aberrant transition from neurogenesis to gliogenesis in DS are recapitulated in our cellular model, and (ii) EVs can mitigate these aberrations through the promotion of the generation of deep and superficial layer neurons and a decrease in the production of astrocytes. Future studies need to explore the beneficial effect of EVs on neurodevelopmental aberrations using earlier time points for intervention.

In the current study, it was also demonstrated that EVs can mitigate pathological phenotypes and alleviate AD pathology in DS-derived CS, including reduction in cell death, decrease in p-tau levels, and alleviation of Aβ40 and Aβ42 depositions. In line with other studies in organoids, we assessed cell death in our CS using CC3 staining as a marker. However, the activation of the executioner caspases does not provide the full picture of the precise cascade of events leading to the induction of the apoptotic process in trisomic cells. Future studies might be required to expose the diverse mechanisms driving the increased expression of CC3 in trisomic CS [87].

The main effect of EVs on Aβ depositions was observed in insoluble fractions and the mechanisms mediating this specific effect of the EVs are still unknown. Insoluble Aβ40 and Aβ42 proteins are present in extracellular amyloid plaques. These insoluble fibrils and soluble oligomers both contribute to cognitive decline in AD. Soluble Aβ species have been shown to be more toxic than insoluble forms of Aβ [88,89]; however, the gradual shift from soluble to insoluble oligomers occurs both at the onset and through the progression of AD [77]. Our findings reveal promising possibilities of therapeutic applications for EVs on the pathological features of AD. Future experiments will be necessary to better understand the mechanism by which EVs are able to alleviate AD pathology in trisomic CS. 

Based on our previous findings showing positive results from EV administration to rhesus monkeys after cortical injury [50,51,52,53], we treated our trisomic CS with the same EVs from the rhesus monkey. The primary targets of EVs are unknown. It is unclear if EVs are directly incorporated into the cells or can indirectly affect endogenous EV signaling between cell types. There is evidence that endogenous EV signaling in DS and AD is disrupted [88,90,91]. Furthermore, it is thought that EV-mediated signaling is one predominant way to spread tau pathology in AD mouse models and in humans [90,92]. Human DS EVs have also been found to contain elevated levels of amyloid proteins and could possibly be a mechanism to transport these proteins to different brain regions [48]. Given that the current study utilizes EVs from the rhesus monkey, future experiments utilizing EVs isolated from either isogenic euploid iPSCs or iPSC-derived progenitors would provide a more physiologically and genetically relevant platform to assess the outcomes and the efficacy of EV administration in trisomic CS. 

The utilization of our CS model of DS to demonstrate the therapeutic effects of EVs is novel and promising. However, there are some limitations that make it difficult to fully understand the extent to which these data precisely translate in vivo. CS lack microglia, a key cell type that plays an important role in the pathophysiology of AD and DS-AD. Indeed, in the context of brain injury, we have shown in rhesus monkeys that EV treatment is associated with a shift from pro-inflammatory to anti-inflammatory microglial morphologies [52]. The lack of microglia in CS limits the translational relevance of our study and the ability to predict whether the EV treatment would be recapitulated in the human DS and DS-AD brain. 

Recent studies have shown that iPSC-derived microglia can be incorporated into spheroids, leading to a more accurate recapitulation of neuroinflammatory phenotypes and electrophysiological properties [93]. The incorporation of microglia into spheroids leads to enhancements in neuronal maturation, synapse formation, and the formation of the neural network as a whole [94]. Recent work from our group has shown the effect of EVs to dampen injury-related hyperexcitability and synapse loss and promote balanced neuronal activity [51]. Future experiments with microglia introduced in the CS would be imperative to allow the testing of potential therapeutics on DS-related inflammatory signature and electrophysiological properties at the synapse level and on a broader network level. This is especially relevant since the maturation of the neurons and astrocytes can be achieved through a more prolonged culturing time period, to allow for the formation of more robust neuronal networks and activity that can be assessed [59,94]. Incorporation of microglia into our CS system would also reveal the effects on tau propagation. Previous work has shown that microglia contribute to the spread of tau proteins via EV secretion, promoting the progression of tauopathy in mice brains and primary cultured murine neurons [90]. Further studies enabling the incorporation of iPSC-derived microglia will illuminate the mechanisms of aberrant DS-related connectivity and the potential efficacy of EVs on neuronal functionality and tau propagation in DS and AD. 

In summary, we demonstrated in human CS the effects of therapeutic EV application on neurodevelopmental phenotypes, including a partial rescue of neuronal deficits in trisomic CS and the potential mitigation of precocious gliogenic shift. Additionally, EVs displayed significant therapeutic effect on cellular phenotypes associated with degenerative changes in DS and were able to mitigate AD pathology by reducing Aβ levels in both the soluble and insoluble fractions, decreasing levels of p-tau concentrations and reducing markers of cell death in trisomic CS. 

## 4. Materials and Methods

### 4.1. Generation of Cortical Spheroids

Human induced pluripotent stem cell (iPSC) lines generated from a 25-year-old woman with DS, including a trisomic WC-24-02-DS-M line and an isogenic euploid control cell line, WC-24-02-DS-B, were used. These lines were produced and validated at Bhattacharyya lab at the University of Wisconsin-Madison and were a kind gift from Dr. Anita Bhattacharyya. These lines were genetically identical apart from the Hsa21 chromosome [95].

IPSCs were routinely passaged and cultured on Matrigel^®^ (cat. 354277 Corning^®^, Corning, NY, USA) using mTeSR™ plus (cat. 85850, StemCell Technologies, Vancouver, BC, Canada). To generate cortical spheroids, cells were washed with PBS (cat. MT21040CV, Fisher Scientific, Waltham, MA, USA), and Accutase (cat. 7920, Stem Cell Technologies) was added for 5 min to dissociate the cells into a single cell suspension. Cells were centrifuged for 5 min at 200 rpm and the pellet was resuspended to a concentration of 15,000 cells/well in 150 mL starter media (mTeSR™ plus supplemented with 50 µM Rock Inhibitor (cat. 12-541-0, Fisher Scientific)). Cells were plated in low adherence V-bottom 96 well plates, to be differentiated further into cortical spheroids (CS) following the protocol described by [58]. 

For the next six days, TeSR5/6 (cat. 5946, StemCell Technologies) media were used with 2.5 µM Dorsomorphin (cat. P5499, Sigma, St. Louis, MO, USA) and 10 µM SB-431542 (cat. S4317, Sigma) and changed daily. On day 7, media were changed to spheroid media containing Neurobasal-A media (cat. 10888022, Life Technologies, Carlsbad, CA, USA) with B-27 supplement (cat. 12587, Life Technologies), GlutaMax (cat. 35050061, Life Technologies), 100 U/mL Penicillin/Streptomycin (cat. 15140122, Life Technologies, 1:100) and Primocin (cat. ant-pm-1, Invivogen, San Diego, CA, USA). From days 7 to 25, 20 ng/mL fibroblast growth factor 2 (FGF2; cat. 233-FB-25/CF, R&D Systems, Minneapolis, MN, USA) and 20 ng/mL epidermal growth factor (EGF; cat. 236-EG-200, R&D Systems) were added. Half of the media was changed every day on days 7–15 and every other day on days 17–25. Spheroids were transferred to ultra-low-attachment 24-well plates with one spheroid per well between days 16–20. 

On day 27, 20 ng/mL brain derived neurotrophic factor (BDNF; cat. AF-450-02, Peprotech, Waltham, MA, USA) and 20 ng/mL neurotrophic factor 3 (NT3; cat. 450-03, Peprotech) were added to spheroid media to induce neural differentiation. Starting on day 27 and continuing for the duration of the experiment, media also included 1% Geltrex (cat. A1569601, Life Technologies). Half of the media’s volume was removed and replaced with fresh media every other day from day 27 to day 39. From day 41 to day 51, spheroid media were not supplemented with any molecules and half of the media was changed every other day. Starting on day 51, 10 ng/mL platelet-derived growth factor AA (PDGF-AA; cat. 221-AA, R&D Systems) and 10 ng/mL insulin-like growth factor (IGF; cat. USA291-GF-200, R&D Systems) were added to spheroid media, and media were changed every two days to expand the oligodendrocyte progenitor cell population. On day 61, 40 ng/mL 3,3′,5-triiodothronine (T3; cat. T6397, Sigma) was added and half the media was changed every two days to direct cells into oligodendrocyte lineage. On and after day 71, spheroid media with no added small molecules were used, changing half the volume every other day.

### 4.2. Extracellular Vesicles and Sample Collection

EVs were isolated from mesenchymal stem cells derived from the bone marrow of young healthy rhesus monkeys, as described previously [50,96]. EVs were administered on day 80, and once per week for the following four weeks. Spheroids were collected at 120 days. 

### 4.3. Cryoprotection and Slice Preparation

Spheroids were fixed overnight in 4% paraformaldehyde (PFA), washed 3 times with PBS, then transferred to 30% sucrose for cryoprotection. Spheroids were embedded in 30% sucrose and optimal cutting temperature embedding medium (OCT) at a 40:60 ratio, frozen at −80 °C and sectioned at 12 μm.

### 4.4. Immunohistochemistry (IHC)

Slides with sectioned spheroids were rehydrated and washed three times with PBST (PBS with 0.1% Triton X-100). The slides were then incubated in blocking solution containing 5% donkey serum for 1 h, followed by incubation with primary antibodies diluted with the blocking buffer overnight at 4 °C. Slides were washed three times in PBST for 10 min each and incubated with secondary antibodies diluted in blocking buffer for 1 h at room temperature, washed three more times for 10 min each, and cover slipped with ProLong^TM^ Gold Antifade Mountant with DAPI (cat. P36931, ThermoFisher, Waltham, MA, USA). 

The following antibodies were used: rabbit anti-CC3 (1:750, cat. 9661-s, Cell Signaling, Danvers, MA, USA); rat anti-GFAP (1:1000, cat. 13-0300, ThermoFisher); rat anti-CTIP (1:500, cat. ab18465, Abcam, Cambridge, UK); mouse anti-SATB2 (1:100, cat. ab51502, Abcam); rabbit anti-TBR1 (1:250, cat. Ab31940, Abcam); mouse anti-Ki67 (1:50, cat. 550609, BD Pharmingen, San Diego, CA, USA); rabbit anti-MAP2 (1:1000, cat. PA5-85755, Life Technologies); goat anti-Sox2 (cat. AF2018, R&D Systems) mouse anti-TUJ1 (1:250, cat. ab14545, Abcam); mouse anti-CC1 (1:150, cat. ab16794, Abcam) and 4G8 (1:1000, cat. 800701, Biolegend, San Diego, CA, USA) and PHF13 (1:200, cat. 829001, Biolegend). All secondary antibodies were Alexa Fluor conjugated (Life Technologies) and were used at the concentration 1:500.

### 4.5. Imaging and Quantification

Each slide contained between two and four spheroids. For each spheroid, three fields were imaged using a Zeiss LSM 710 confocal microscope system (Carl Zeiss AG, Oberkochen, Germany). For cellular markers, image stacks of each field were acquired using a 20× 0.75 N.A. objective lens at 0.277 × 0.277 × 1 µm (z-step) voxel size resolution. Sections containing developing neuron and NPC markers (SATB2, CTIP2, TBR1, Ki67, and SOX2) were chosen in areas surrounding the VZ-like structures in the CS, identified using morphology and presence of positive markers. For markers of astrocytes and cell death, random edge regions were selected. Quantification of images was performed using manual counting on ImageJ (version: 2.9.0/1.54b, U. S. National Institutes of Health, Bethesda, Maryland, USA) particle analysis. Particle analysis was performed by selecting the sections of the Z stack containing the majority of the cells, splitting the image into separate channels, and setting the threshold for each channel separately. Percent area for each image was averaged between slices in the z stack, then averaged again with all images from a single spheroid. Then, the average from each spheroid was averaged further by treatment group and reported as mean ± standard error of the mean. 

Additionally, we used the ACEq (available for download at https://www.bumc.bu.edu/anatneuro/ella-zeldich-lab/, accessed on 8 December 2022) counting app developed in our lab, as previously described in [70], for automatic cell counting of nuclear markers. ACEq was used to quantify SATB2, CTIP2, TBR1, Ki67, and SOX2 nuclear stainings normalized to DAPI. The cell counts from analyzing each image using the ACEq app were normalized to DAPI cell counts per specific image, the counts for each spheroid were averaged, and the mean result for each treatment group was quantified. Results are reported as mean ± standard error of the mean as a percentage of the euploid group. The two outliers for SATB2, CTIP2, and TBR1 stainings for the first differentiation experiment were removed based on the results of the interquartile range (IQR) × 1.5 outlier test.

In addition to these images acquired for quantification, we acquired additional images of specific areas and structures within our sampled fields just for the qualitative visualization of labeling patterns. To visualize the labeling patterns of cellular markers, we used a 20× 0.75 N.A. objective with 2.5 to 3.5× zoom to image specific areas of interest at 0.1 × 0.1 × 1 µm voxel size resolution. To visualize labeling patterns of the subcellular marker tau, we imaged z-stacks of fields using a 40× 1.0 N.A. objective at 0.1 × 0.1 × 1 µm voxel resolution.

### 4.6. Fractionation of Spheroids and Aβ Enzyme-Linked Immunosorbent Assay (ELISA)

Four spheroids were pooled together per sample, flash frozen and stored at −80 °C. For whole homogenates, samples were lysed in RIPA buffer (50 mM Tris pH 8, 150 mM NaCl, 0.5 mM EDTA, 1% NP-40, 0.1% sodium deoxycholate and 0.1% SDS) (cat. 89900, ThermoFisher) with protease inhibitor cocktail (1:100, cat. 87785, ThermoFisher) and phosphatase inhibitor cocktail (1:100, cat. 78420, ThermoFisher) added [97]. For fractionation, samples were sonicated and ultracentrifuged at 42 g for 1 h at 4 °C. The soluble fraction was present in the liquid phase and was separated, leaving only the insoluble fraction in the pellet. Sixty microliters of urea buffer (8 M Urea; 2 M Thiourea; 4% CHAPS; 30 mM Tris HCl pH 8.5) [97] was added to the insoluble pellet and samples were sonicated again. Samples were stored at −80 °C until ready to be used. ELISA was performed using Aβ40 and Aβ42 ELISA kits (cat. KHB3481, Life Technologies), following the manufacturer’s protocol, on both the soluble and insoluble fractions. Total protein concentration was measured using the bicinchoninic acid (BCA) method and ELISA results for Aβ40 and Aβ42 were normalized to these concentrations.

### 4.7. Western Blotting

Whole CS homogenates were used for Western blotting. Protein samples were prepared by adding 6 µL 4× Laemni buffer (cat. 1610747, BIO-RAD, Hercules, CA, USA) containing 10% β-mercaptoethanol to 18 µL of a solution containing 30 µg total protein in whole homogenates and water. Samples were boiled at 95 °C for 5 min and separated by sodium dodecyl sulphate-polyacrylamide gel electrophoresis using precast 4–20% Mini-PROTEAN TGX gels (cat. 4561093, BIO-RAD). Samples were then transferred to a nitrocellulose membrane using Trans-Blot Turbo Transfer Pack per manufacturer instructions (cat. 1704158, BIO-RAD). The membrane was blocked for 30 min in EveryBlot Blocking Buffer (cat. 12010020, BIO-RAD) at room temperature with shaking and incubated overnight at 4 °C with the primary antibody in EveryBlot Blocking Buffer. The following antibodies were used: mouse anti-PHF13 (1:1000, cat. 829001, Biolegend); mouse anti-actin (1:10,000, cat. MAB1501, Millipore Sigma, Burlington, MA, USA)

After three washes with TBST, the membrane was incubated in KPL peroxidase-conjugated mouse IgG secondary antibody (1:5000, cat. 5220-0341, Seracare, Milford, MA, USA) in EveryBlot Blocking Buffer for 1 h at room temperature with shaking. The membrane was washed two times in TBST and once in TBS and detection was performed using Azure 300 (Azure Biotech, Houston, TX, USA). Gel analysis was performed using ImageJ software to quantify relative protein expression levels normalized to actin by densitometry. Western blots for three independent differentiation experiments were used for quantification.

### 4.8. Measuring Spheroid Diameter

Spheroid diameters were measured in real time on day 30, 50, 80, and 120 using a Swiftcam 10 Megapixel Microscope Digital Camera and Imaging software at 4× magnification. 

### 4.9. Data Analysis

Quantitative data are expressed as means ± SEM. Comparisons between experimental groups were analyzed using a two-tailed, paired *t*-test. ANOVA was used for multi-group data comparisons followed by Tukey’s post hoc test to assess differences between conditions. Statistical analyses were performed with GraphPad Prism 9.0 (GraphPad Software Inc., San Diego, CA, USA). Values of *p* < 0.05 were considered significant.

## Figures and Tables

**Figure 1 ijms-24-03477-f001:**
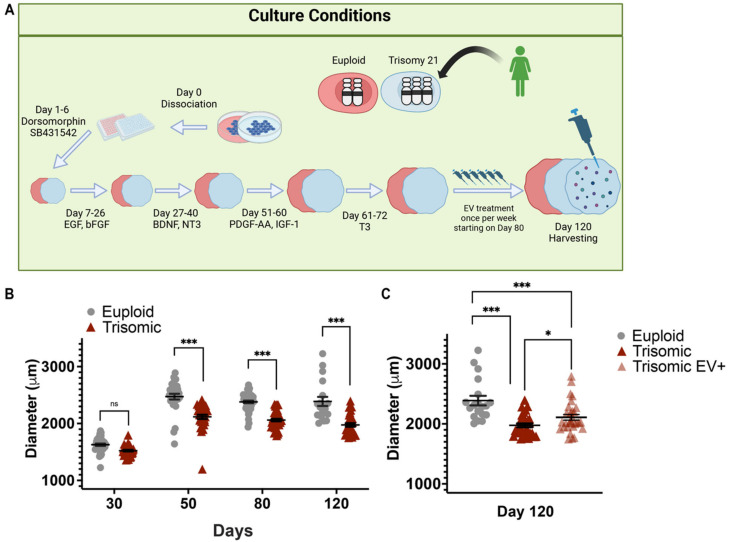
EV treatment preserved the size of trisomic CS. (**A**) Schematic representation of CS development protocol and experimental design demonstrating differentiation of iPSCs into CS. Created with BioRender.com (accessed on 14 November 2022). (**B**) Vertical scatter plot showing individual data (closed gray circles and red triangles) and mean +/− SEM (black lines) of measured diameters (µm) of euploid and trisomic CS generated from isogenic cell lines on days 30 (euploid n = 38; trisomic n = 37), 50 (euploid n = 31; trisomic n = 35), and 80 (euploid n = 45; trisomic n = 44). (**C**) CS from day 120 showing euploid (n = 38) and trisomic groups with or without EV treatment (trisomic n = 73; trisomic EV+ n = 75). The graphs show the results obtained from the data collected from three independent experiments. Statistical analysis was performed by one-way ANOVA comparing each group’s mean. Results are shown as mean ± SE.* *p* < 0.05, *** *p* < 0.001, ns = non-significant.

**Figure 2 ijms-24-03477-f002:**
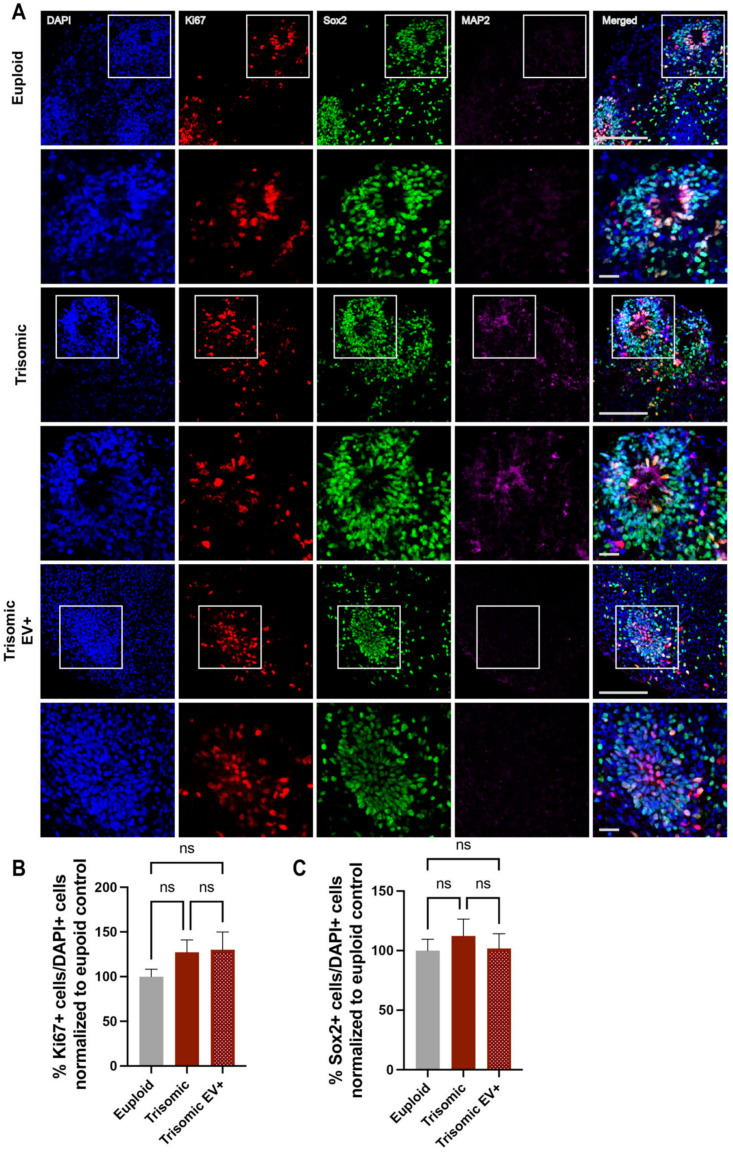
Measurements of proliferative markers in CS are unaffected by EV treatment. (**A**) Single and merged channels of confocal images showing DAPI staining (blue) with IHC labeling of MAP2 (magenta), SOX2 (green), and Ki67 (red) in euploid and trisomic untreated and EV-treated CS generated from isogenic cell lines showing VZ-like structures. The top panels show low magnification images, scale bar = 100 µm. White boxes represent high magnification insets seen in the bottom panels, scale bar = 20 µm. (**B**,**C**) Bar graphs showing quantification of the percentages of SOX2+ and Ki67+ cells in euploid CS and trisomic untreated and EV-treated CS, quantified as the ratio of the numbers of SOX2+ and Ki67+ to the number of all cells marked with DAPI and normalized to the euploid control that was calculated as 100% for each independent experiment. Cells were counted using the ACEq counting app; euploid n = 20; trisomic n = 23; trisomic EV+ n = 25 CS per group. These graphs reflect the results obtained from data collected from three independent experiments. Statistical analysis was performed via unpaired *t*-test with Welch’s correction comparing all three groups to each other. Results are shown as mean ± SE, ns = non-significant.

**Figure 3 ijms-24-03477-f003:**
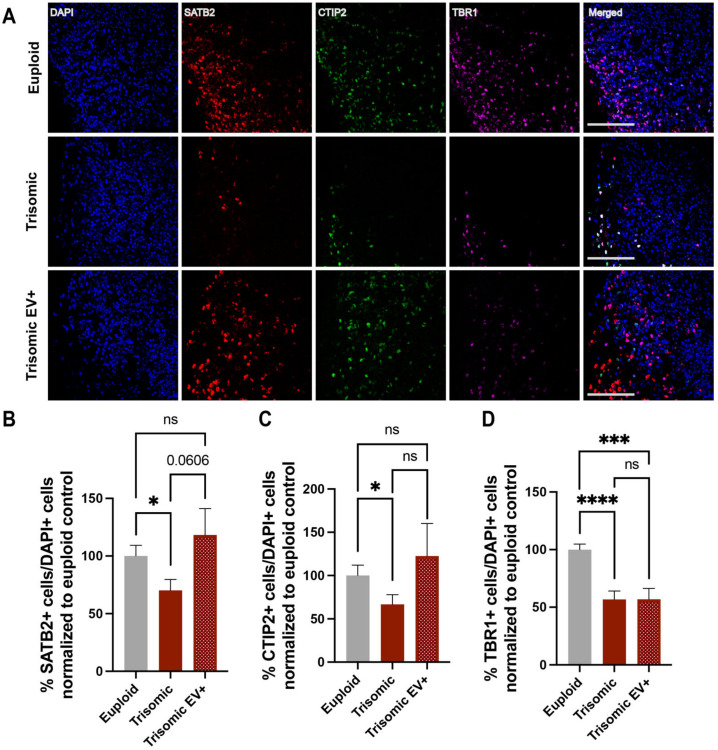
The effect of EV treatment on neuronal populations in trisomic CS. (**A**) Single and merged channels of confocal images showing DAPI staining with IHC labeling of TBR1 (magenta), CTIP2 (green) and SATB2 (red) in euploid and trisomic untreated and EV-treated CS generated from isogenic cell lines. Scale bar = 100 µm. (**B**–**D**) Bar graphs showing quantification of percentages of SATB2+, CTIP2+, and TBR1+ cells in euploid CS and trisomic untreated and EV-treated CS, quantified as the ratio of the numbers of CTIP2+, SATB2+, and TBR1+ to the number of all cells marked with DAPI and normalized to the euploid control that was calculated as 100% for each independent experiment. Cells were counted using the ACEq counting app; euploid n = 18; trisomic n = 22; trisomic EV+ n = 22 CS per group. These graphs reflect the results obtained from data collected from three independent experiments. Statistical analysis performed via unpaired *t*-test with Welch’s correction comparing all three groups to each other. Results are shown as mean ± SE. * *p* < 0.05, *** *p* < 0.001, **** *p* < 0.0001, ns = non-significant.

**Figure 4 ijms-24-03477-f004:**
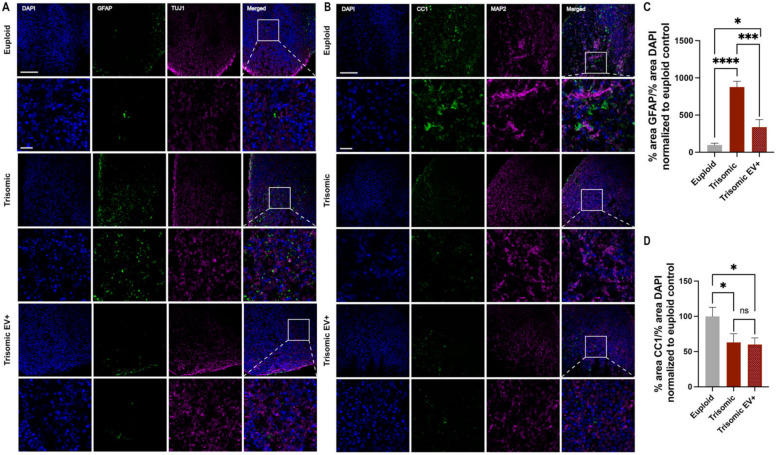
The effect of EV treatment on astrocyte and oligodendrocyte populations in trisomic CS. (**A**) Single and merged channels of confocal images showing DAPI staining (blue) with IHC labeling of GFAP (green) and TUJ1 (magenta) in euploid and trisomic untreated and EV-treated CS generated from isogenic cell lines. (**B**) Single and merged channels of confocal images showing DAPI staining (blue) with IHC labeling of CC1 (green) and MAP2 (magenta) in euploid and trisomic untreated and EV-treated CS generated from isogenic cell lines. Top panels in A and B show low magnification images, scale bar = 100 µm. White boxes represent high magnification insets in bottom panels, scale bar = 20 µm. (**C**) Bar graphs showing the quantification of the percentage of GFAP expression and (**D**) the percentage of CC1 expression in euploid CS and trisomic untreated and EV-treated trisomic CS. The percentage of each marker was quantified as the ratio of the area occupied by the label (GFAP or CC1) to the area occupied by DAPI in each image and normalized to the euploid control that was calculated as 100% for each independent experiment. The quantification was performed through particle analysis via ImageJ; euploid, n = 18; trisomic, n = 22; trisomic EV+ n = 22 CS per group. These graphs display results from three independent experiments. Statistical analysis was performed via unpaired *t*-test with Welch’s correction comparing all three groups to each other. Results are shown as mean ± SE. * *p* < 0.05, *** *p* < 0.001, **** *p* < 0.0001, ns = non-significant.

**Figure 5 ijms-24-03477-f005:**
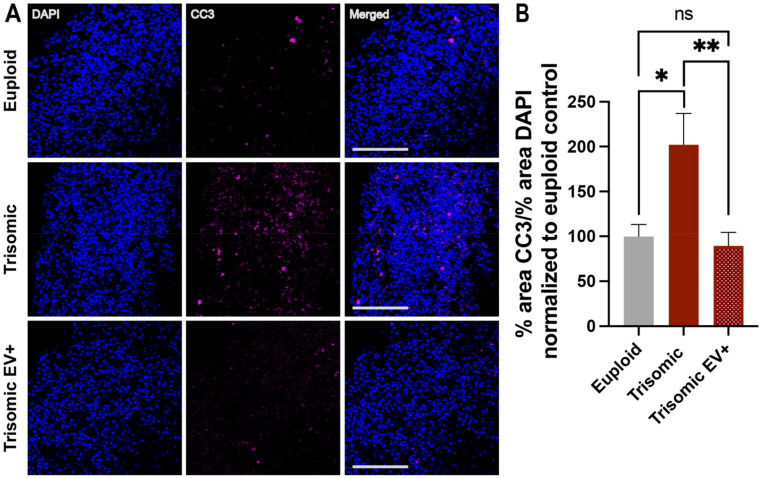
The extent of cell death is reduced in trisomic CS treated with EVs. (**A**) Single and merged channels of confocal images showing DAPI staining with IHC labeling of CC3 (magenta) in euploid and trisomic untreated and EV-treated CS generated from isogenic cell lines. Scale bar = 100 µm. (**B**) Bar graph showing the quantification of percentage of CC3 expression in euploid CS and trisomic untreated and EV-treated CS, quantified as the ratio of the area occupied by CC3 to the area occupied by DAPI and then normalized to the euploid control that was calculated as 100% for each independent experiment. The quantification was performed through particle analysis via ImageJ; euploid n = 18; trisomic n = 22; trisomic EV+ n = 22 CS per group. These graphs display results from three independent experiments. Statistical analysis performed via unpaired *t*-test with Welch’s correction comparing all three groups to each other. Results are shown as mean ± SE.* *p* < 0.05, ** *p* < 0.01, ns = non-significant.

**Figure 6 ijms-24-03477-f006:**
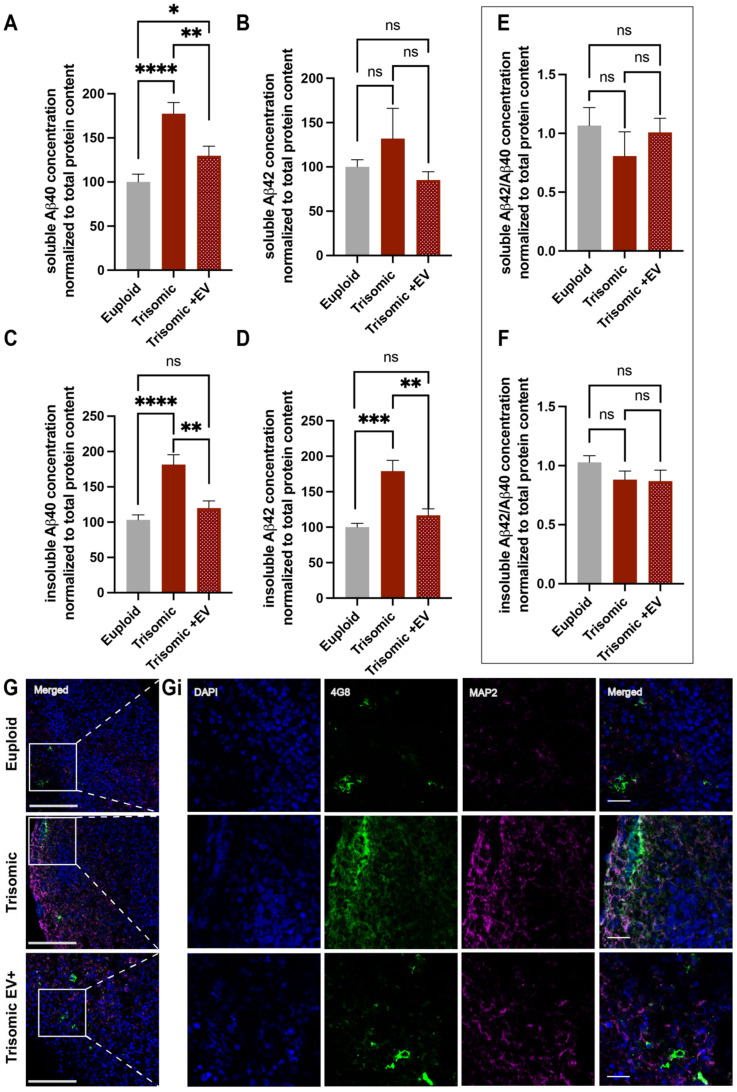
Measurements of Aβ40 and Aβ42 levels following EV treatment. (**A**,**B**) Bar graphs of soluble Aβ40 (**A**) and Aβ42 (**B**) concentrations and; (**C**,**D**) insoluble Aβ40 (**C**) and Aβ42 (**D**) concentrations, calculated as a ratio of Aβ40 and Aβ42 to total protein content assessed via ELISA in euploid and trisomic untreated and EV-treated CS and normalized to the euploid control that was determined as 100% for each independent experiment; (**E**,**F**) Bar graph of ratio of soluble (**E**) and insoluble (**F**) Aβ42/Aβ40 concentration normalized to total protein content in euploid and trisomic untreated and EV-treated CS, and normalized to the euploid control that was determined as 100% for each independent experiment. For Aβ40, euploid n = 12; trisomic n = 14; trisomic EV+ n = 15 samples per group generated from 4–5 CS combined together per sample. For Aβ42, euploid n = 8; trisomic n = 10; trisomic EV+ n = 11 samples per group generated from 4–5 combined together per sample. These graphs display results from three independent experiments. Statistical analyses were performed via unpaired *t*-test with Welch’s correction comparing all three groups to each other. Results are shown as mean ± SE. * *p* < 0.05, ** *p* < 0.01, *** *p* < 0.001, **** *p* < 0.0001, ns = non-significant. (**G**,**Gi**) Low magnification merged channel images. White boxes represent insets seen in Gi showing single and merged channels of confocal images of high magnification insets with IHC labeling of 4G8 (green), MAP2 (magenta), and DAPI in euploid and trisomic untreated and EV-treated CS. Low magnification scale bar = 100 µm. High magnification inset scale bar = 20 µm.

**Figure 7 ijms-24-03477-f007:**
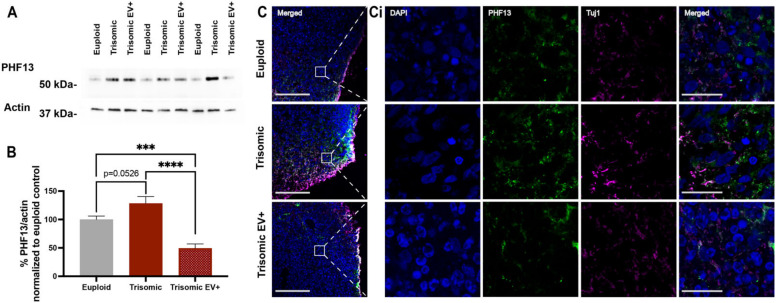
Measurements of PHF13 levels in EV-treated CS. (**A**) Representative Western blot image of tau phosphorylation visualized with PHF13 antibody in whole homogenates of euploid, trisomic, and EV-treated trisomic CS. (**B**) Bar graph of densitometric analysis of the content of phosphorylated tau quantified as a ratio of PHF13 intensity over actin intensity in CS and normalized to the euploid control that was calculated as 100% for each independent experiment; euploid n = 12; trisomic n = 12; trisomic EV+ n = 5 samples per group generated from 4–5 CS combined together per sample. These graphs display results from three independent experiments. Statistical analyses were performed via unpaired *t*-test with Welch’s correction comparing all three groups to each other. Results are shown as mean ± SE. *** *p* < 0.001 **** *p* < 0.0001. (**C**,**Ci**) Low magnification merged channel images. White boxes represent high magnification insets seen in (**Ci**) showing single and merged channels of confocal images with IHC labeling of PHF13 (green), Tuj1 (magenta), and DAPI in euploid and trisomic untreated and EV-treated CS. Low magnification scale bar = 100 µm. High magnification inset scale bar = 20 µm.

## Data Availability

Not applicable.

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
