# Peer review of "Extracellular Vesicle Treatment Alleviates Neurodevelopmental and Neurodegenerative Pathology in Cortical Spheroid Model of Down Syndrome"

_ijms, 2023, doi:10.3390/ijms24043477_

Round 1

Reviewer 1 Report

Extracellular Vesicle treatment alleviates neurodevelopmentaland neurodegenerative pathology in cortical spheroid model of Down Syndrome

Comments 

1、Figures 3, 4, 6, and 7 have "****", but only in Figure 7 (in Line 441) is there a comment. There are no comments in lines 276, 323, and 412.The format of "Pvalue" in all comments is wrong.

2. In Figure 7, the WB results are very unclear and not convincing evidence.

3. In Figure 2 (in Lines 218-219), the scale is described as“The top panels show low magnification images, scale bar = 100μm (20x lens with zoom 1.5). The bottom panels show insets at higher magnification, scale bar = 20μm (20x lens with zoom 2.5).”. In Figure 4 (in Lines 314-315), the scale is described as“Top panels in A and B show low magnification images, scale bar = 100μm (20x lens with zoom 1.0). Bottom panels in A and B show insets at higher magnification, scale bar = 20μm (20x lens with zoom 3.5).”.In Figure 6 (in Lines 218-219), the scale is described as“Low magnification scale bar = 100μm (20x lens with zoom 1.5). Higher magni- fication inset scale bar = 20μm (20x lens with zoom

3.5)”.

In Figure 7 (in Lines 433-444), the scale is described as“The scale bar is 100μm (20x lens with zoom 1.5) and 20μm in inset (40x lens with zoom 4.5).”. Why don't the photos have the same magnification?Why not use a 40x lens for shooting in Figures 2, 4 and 6?

4.In IHC, I observed that the expression of fluorescence in the figure was not uniform, for example, Figure 6 (G),Figure 7 (C). On what criteria did the authors base their selection? Would the conclusions have changed if other horizons had been chosen?

5.The evidence provided in the article is dominated by IHC, with very little other validation, and is less convincing.Is there more evidence to support the conclusions in the article?

6. There are errors in wording and formatting in the article.

Author Response

We thank the Reviewer for the helpful and insightful comments and provide our responses to each one of the comments.

1、Figures 3, 4, 6, and 7 have "****", but only in Figure 7 (in Line 441) is there a comment. There are no comments in lines 276, 323, and 412.The format of "Pvalue" in all comments is wrong.

Response: The thank the reviewer for noticing our oversight. We have updated the figure legends in the appropriate figures to include ****p<0.0001.

  1. In Figure 7, the WB results are very unclear and not convincing evidence.

Response: We agree with the reviewer that the WB results can be presented better. We repeated the WB experiment and are showing now the results of the triplicates obtained in spheroids homogenates generated through independent differentiation experiments.

We also added a supplementary file showing the raw image of the WB.  Please note that the quantification results were made on multiple samples (collected through independent experiments and the numbers of samples are indicated in the legends).

  1. In Figure 2 (in Lines 218-219), the scale is described as“The top panels show low magnification images, scale bar = 100μm (20x lens with zoom 1.5). The bottom panels show insets at higher magnification, scale bar = 20μm (20x lens with zoom 2.5).”. In Figure 4 (in Lines 314-315), the scale is described as“Top panels in A and B show low magnification images, scale bar = 100μm (20x lens with zoom 1.0). Bottom panels in A and B show insets at higher magnification, scale bar = 20μm (20x lens with zoom 3.5).”.In Figure 6 (in Lines 218-219), the scale is described as“Low magnification scale bar =100μm (20x lens with zoom 1.5). Higher magni- fication inset scale bar = 20μm (20x lens with zoom 3.5)”.

In Figure 7 (in Lines 433-444), the scale is described as“The scale bar is 100μm (20x lens with zoom 1.5) and 20μm in inset (40x lens with zoom 4.5).”. Why don't the photos have the same magnification?Why not use a 40x lens for shooting in Figures 2, 4 and 6?

Response: We chose the objectives and zoom level based on the specific structures we imaged and the purpose of the imaging (i.e. for quantification or for qualitative viewing only). 

For all images used for quantification, we used the 20x 0.75 N.A. objective (at full field of view, which is 1.5x zoom in the Zeiss 710) to acquire confocal z-stacks from 3 fields per organoid, at a voxel resolution of 0.3 x 0.3 x 1 µm, to count cellular markers.

After imaging these fields for quantification, we then took additional “zoomed in” higher magnification images of specific areas to show examples of labeled structures. These images were not quantified and were just for visualization. We chose the objective and zoom level based on the structure we wanted to capture.  The 20x gives us a bigger field of view than the 40x, and thus we needed to use this objective to image the large structures in Figure 2, 4, 6. For Figure 7, we needed to image Tau, which is a subcellular structure, and we did not need a big field of view, but we need higher optical sampling, thus we used the 40x 1.0 NA objective. Based on Abbe’s diffraction limit, which states that resolution is based on the numerical aperture (NA), a 20x 0.75 N.A. and 40x 1.0 N.A. are very equivalent in terms of optical resolution. Thus, the field of view that is the biggest difference between these two objectives. Indeed, the final digital resolution of images acquired with a 20x 0.75 NA objective with 2-3x zoom, and 40x 1.0 NA objective are equivalent (about 0.1 x 0.1 x 1 µm).

We apologize for the confusion with our imaging parameters. It is important to note that according to the principles of microscopy and Abbe’s equation, it is really the objective Numerical Aperture (N.A.) and the final voxel resolution that matters for what an image can resolve and visualize. Hence, we have now added details to state the N.A. of the objectives and the precise voxel resolution used for all acquired images in Methods for clarity. To avoid confusion, we also have deleted information of the zoom and objective configurations in the legends and described this in the context of our detailed imaging workflow and parameters in the Methods.

4.In IHC, I observed that the expression of fluorescence in the figure was not uniform, for example, Figure 6 (G),Figure 7 (C). On what criteria did the authors base their selection? Would the conclusions have changed if other horizons had been chosen?

Response: The images used in Figure 6 and Figure 7 are representative of the results obtained through Western Blot and ELISA and are not used for the quantification. The quantification of the results shown in figures 6 and 7 is done based on the ELISA and WB, respectively. While the reviewer is correct, and the fluorescence is not entirely uniform through each edge (horizon) of the imaged CS, this situation is also comparable to the human DS-AD brain where the pathological depositions are not distributed uniformly. That being said, we routinely image several edges of our spheroids and the images we are showing are representative of other edges that we image.

5.The evidence provided in the article is dominated by IHC, with very little other validation, and is less convincing. Is there more evidence to support the conclusions in the article?

Response: Along with IHC, CS measurements, Western Blot and ELISA were used to support our conclusions about size, hyperphosphorylated tau and amyloid beta depositions in trisomic CS.  

We would like to note that the generation of CS and quantification of IHC results were done up to the highest standards. We used three differentiation experiments, which means that every time the new CS were generated from iPS cells and cultured in individual wells for 120 days.

We collected 8-15 organoids from each independent differentiation experiment (the exact numbers are indicated in the legends) that were stained and each organoid was imaged from 3 different randomly chosen edges.  The results were quantified as we described in detail in our methods section. Then we averaged the data from the imaged areas to generate the data per organoid and the mean value for each independent experiment was quantified. IHC is heavily validated and extensively applied methodology used by different groups in a scientific community. We feel that this methodology supported by our stringent approaches provides a solid basis for the conclusions we draw in the manuscript.

  1. There are errors in wording and formatting in the article.

Response: We used grammar editing software (Grammarly) to edit our manuscript. We also noted that the figures uploaded through the original submission were exported as low-resolution images. Now, we exported all the images preserving high resolution and incorporated them into our resubmission material.

Reviewer 2 Report

This is a very interesting paper in which authors demonstrated the therapeutic effect of Extracellular Vescicles (EVs) derived from bone-marrow Mesenchymal Stromal cells in a cortical spheroid (CS) model of Down Syndrome.

Authors first demonstrated that EV-treated trisomic CS preserved the volume compared to the untreated CS; after that they investigated the potential processes that could promote this preservation of volume such as proliferation, neurogenesis, and processes related to cell death and cell viability.

In my opinion, authors must improve the quality of the pictures in all figures, also in the supplementary files. Moreover, in figure 7 the quality of the western blot picture is very low and in the supplementary file there isn’t the row picture, but the same image in the same low quality.

In addition, authors investigated the cell death performing a detection of Cleaved Caspase 3. However, apoptosis relies on an intracellular proteolytic cascade which is essentially mediated by two types of caspases, namely initiator caspases (such as caspase-8 and caspase-9) and executioner caspases (caspase-3, caspase-6 and caspase-7). Two important mammalian pathways which can activate an initiator caspase are the extrinsic pathway and the intrinsic (or mitochondrial) pathway.

Notably, the intrinsic pathway of apoptosis can be activated following oxidative damage to mitochondrial proteins, DNA damage and peroxidative damage to mitochondrial membrane lipids driven by excessive levels of reactive oxygen species (ROS) and reactive nitrogen species (RNS)(PMID:29052145). On this premises, authors should evaluate the cell death performing another cell viability assay.

Author Response

We thank the Reviewer for the helpful and insightful comments and provide our responses to each one of the comments.

In my opinion, authors must improve the quality of the pictures in all figures, also in the supplementary files. Moreover, in figure 7 the quality of the western blot picture is very low and in the supplementary file there isn’t the row picture, but the same image in the same low quality.

 Response: The reviewer is correct and indeed, the figures uploaded through the original submission were exported as low-resolution images. We are thankful to the reviewer for noticing our oversight.

 Now, we exported all the images preserving high resolution and incorporated them into our resubmission material.

We also agree with the reviewer that the WB results can be presented better. We repeated the WB experiment and are showing now the results of the triplicates obtained in spheroids homogenates generated through independent differentiation experiments.

We also added a supplementary file showing the raw image of the WB.  Please note that the quantification results were made on multiple samples (collected through independent experiments and the numbers of samples are indicated in the legends).

In addition, authors investigated the cell death performing a detection of Cleaved Caspase 3. However, apoptosis relies on an intracellular proteolytic cascade which is essentially mediated by two types of caspases, namely initiator caspases (such as caspase-8 and caspase-9) and executioner caspases (caspase-3, caspase-6 and caspase-7). Two important mammalian pathways which can activate an initiator caspase are the extrinsic pathway and the intrinsic (or mitochondrial) pathway.

Notably, the intrinsic pathway of apoptosis can be activated following oxidative damage to mitochondrial proteins, DNA damage and peroxidative damage to mitochondrial membrane lipids driven by excessive levels of reactive oxygen species (ROS) and reactive nitrogen species (RNS)(PMID:29052145). On this premises, authors should evaluate the cell death performing another cell viability assay.

Response: We agree with the reviewer that the evaluation of the cleaved caspase 3 (CC3)  represents the merge-point of the intrinsic and extrinsic apoptotic pathway that can be initiated by diverse stressors and mechanisms. However, the mechanism of cell death in trisomic spheroids was outside of the scope of our work and we assessed the expression of CC3 as a general assessment of cell death in trisomic CS in the presence and absence of EV treatment. CC-3 is a widely used marker for cell death evaluations in fixed human organoids as has been shown through numerous other publications (Contreras et al, 2023, Qian et al, 2016, Qian et al, 2020, Mesci et al, 2022, Li et al, 2022). To address reviewer’s comment, we added the potential limitation of CC-3 assessment to the discussion (lines 542-547) .

Round 2

Reviewer 1 Report

can be accepted in the present form 

Reviewer 2 Report

Thank you for your replies.